

# Application of public emotion feature extraction algorithm based on social media communication in public opinion analysis of natural disasters

Shanshan Li and Xiaoling Sun

Institute of Disaster Prevention, School of Information Engineering, Langfang, Hebei, China

## ABSTRACT

Natural disasters are usually sudden and unpredictable, so it is too difficult to infer them. Reducing the impact of sudden natural disasters on the economy and society is a very effective method to control public opinion about disasters and reconstruct them after disasters through social media. Thus, we propose a public sentiment feature extraction method by social media transmission to realize the intelligent analysis of natural disaster public opinion. Firstly, we offer a public opinion analysis method based on emotional features, which uses feature extraction and Transformer technology to perceive the sentiment in public opinion samples. Then, the extracted features are used to identify the public emotions intelligently, and the collection of public emotions in natural disasters is realized. Finally, through the collected emotional information, the public's demands and needs in natural disasters are obtained, and the natural disaster public opinion analysis system based on social media communication is realized. Experiments demonstrate that our algorithm can identify the category of public opinion on natural disasters with an accuracy of 90.54%. In addition, our natural disaster public opinion analysis system can deconstruct the current situation of natural disasters from point to point and grasp the disaster situation in real-time.

## INTRODUCTION

Social media is the main way people obtain news, express opinions, share information and exchange emotions (*Jia et al., 2017*). When natural disasters occur, social media has become one of the leading platforms for the public to express their feelings and discuss the event. Therefore, through the analysis of the content on social media, the emotional characteristics of the public to natural disaster events can be extracted, which can provide helpful reference information for disaster emergency response and follow-up work (*Wang & Wu, 2022*).

Natural disasters severely impact the public's lives, which may lead to casualties, property losses, and threats to life and property safety. Hence, they often trigger strong emotional responses from the public. The public's emotional reactions and attitudes often impact disaster emergency response and follow-up work. People often learn about

Corresponding author
Shanshan Li, liss@cidp.edu.cn

disasters and express their feelings and attitudes through various channels. As one of the main channels for the public to express their feelings, social media carries a large amount of information and emotional expressions. Analyzing public emotions on social media can effectively understand the public's attitude and emotional reaction to natural disasters (*Wang, Wang & Li, 2022*; *Zhou, Tian & Zhong, 2022*). It mainly focuses on three aspects: (a) The public emotional response will affect the spread and evolution of public opinion. The emotional reactions and attitudes of the public in natural disasters are often expressed and transmitted through social media and other channels, which will impact the dissemination and evolution of public opinion (*Houston et al., 2015*). (b) Public emotional reactions can affect disaster emergency response. Public emotional reaction and attitude often have an essential impact on disaster emergency response, and the public's emotional reaction and attitude have a certain impact on the decision-making and actions of the government and relief agencies (*Yigitcanlar & Kankanamge, 2023*). (c) Public emotional response can provide reference information for disaster management. Analyzing public emotions on social media makes it possible to understand the public's concerns and emotional expressions during natural disasters. It also provides helpful reference information for the government and relief agencies to manage disasters better (*Karimiziarani et al., 2022*).

New media based on the network platform is increasingly popular, especially online media, the most powerful platform for people to communicate and express their emotions daily. People express their inner emotions through speech, audio, or video on social media (*He et al., 2017*). As the most significant way of describing public opinion in social media, the text is highly concise, concise and comprehensive, which can fully arouse the resonance of the public and drive the direction of people's thoughts. Text emotion recognition refers to the analysis of the text to determine if the text includes emotions. Emotion recognition is the foundation and premise of emotion classification. Among the massive amount of social information, there are too much emotional information and non-emotion information. Among them, emotionless captions are an accurate description of individuals. Inspirational texts often include personal information, such as happiness, anxiety, surprise, *etc*. The main objects of text analysis are emotional words. So, it is crucial to recognize emotion types in massive social texts, and they can even directly affect the direction of public opinion. Text emotion recognition can narrow down the object of emotion analysis and boost emotion classification (*Cao, 2022*).

Natural disasters have the characteristics of being sudden and rapid. The public emotions in natural disasters are widely expressed through social media and mixed with a massive amount of other information, making it difficult to find. Considering the above characteristics, we urgently need to accurately find public emotional information from the internet. In addition, in natural disasters, the public's emotions are complex and contain numerous emotional information. Existing methods are difficult to accurately identify public sentiment vectors in a large amount of data streams, which can easily cause secondary damage to public sentiment. To ensure that the model can quickly and accurately locate public emotional information and accurately identify it, this article explores the application of social media in analyzing public opinion in natural disasters. By

proposing a public emotion feature extraction algorithm based on social media transmission, the collection and control of public opinion during the disasters can be realized to quickly guide the distribution of materials, post-disaster reconstruction and other social problems, which can provide reasonable technical support for social governance. Main contributions of our article are as follows:

1) We apply the extracted feature and Transformer to propose a public opinion analysis method, which can perceive the sentiment.
2) We propose a Public Emotion Feature extraction algorithm to enhance the social media communication.
3) We apply the public emotional information to intelligently recognize the state of social media communication.

# RELATED WORKS

## Emotion recognition of public opinion text

Text emotion recognition is extracting the text feature information with emotional color and emotional tendency from the original data using specific text mining methods to identify. The general process of text emotion recognition is as follows: first, the actual words are quantified as word embedding vectors. Then, a deep neural network is constructed to achieve the deep emotional semantic features of the text. Finally, the classifier is used to complete the recognition and classification of emotions. Compared with the traditional machine learning algorithm for text emotion recognition, deep learning has been gradually replaced and employed for text emotion recognition.

In 2015, *Khatri, Singhal & Johri (2014)* constructed a single neural network (NN) model for text emotion analysis and achieved good experimental results. However, the recognition accuracy will be reduced due to complex emotional text features. In 2016, *Poria et al. (2016)* used convolutional neural network (CNN) to obtain information on the emotional text. They input a 306-dimensional vector transformed from each word to the model and the convolution operators to extract features, which achieved good experimental results. In the same year, *Wang et al. (2016)* used the regional CNN-LSTM model to analyze the emotional dimension. Unlike the traditional CNN, which takes the entire text as input, the input text was divided into multiple regions, and the useful emotional information in each region was extracted. The long short-term memory network (LSTM) serially integrates this regional information into different regions.

Experiments demonstrate that the proposed method surpasses the traditional emotion recognition methods proposed by predecessors. In 2017, *Liu, Qiu & Huang (2017)* proposed an adversarial multi-task learning framework. This model framework solves the interference problem caused by task-specific features when extracting shared features. It effectively avoids the interference between the sharing and private layers to boost the classification of text emotion effectively. In 2018, *Su et al. (2018)* applied the LSTM to propose a text emotion recognition method. In this method, they employ the Word2vec model to achieve the semantic word vector for each word in the input text. Additionally,

they map each lexical word to all sentiment words defined in the sentiment dictionary and obtain a sentiment word vector. Then, the autoencoder features are connected to the semantic word features to form the features of the final text. Finally, based on the given text feature of the whole sentence, the context emotion evolution is determined by LSTM. Finally, based on the sequence of text features of the entire sentence, LSTM is employed to model the context emotion of the input text. In 2019, *Zhang & Zhou (2019)* proposed a text emotion classification based on the fusion of CNN and Bi-LSTM, which used the Bi-LSTM network to extract the global emotional feature information in the text more effectively. *Chen et al. (2022)* research the analysis of the public opinion fighting by the emotional text in Chinese Weibo about the Russia–Ukraine war. *Peng et al. (2022)* conduct the survey about the textual emotion and summarize the deep learning methods and traditional methods. *Majeed et al. (2022)* build a *corpus*, namly Deep-EmoRU, which can conduct the emotional polarity of the Roman Urdu text.

## Research on intelligent analysis method of public opinion

Danmaku data mining is to mine the information people pay attention to from many danmaku data, analyze the data characteristics, and obtain the corresponding results. A web crawler is a vital tool for getting danmaku data. To address the issue that the amount of danmaku data generated in the crawling process is large, but the capacity of the danmaku pool is limited, *Ye & Zhao (2019)* used a Loop Adaptive algorithm based on Hot Spot Detection (LAABHD) to solve the problem that danmaku pool is discarded when it has type capacity. *Witten & Frank (2002)* introduced the techniques and related tools of data mining and machine learning and provided a set of Weka toolkits for mining and analyzing data. As a new type of comment data, danmaku text contains many spoken expressions. Famous words and emojis are also common in danmaku text. Compared with traditional comment data, danmaku text requires more detailed data-cleaning operations. *Yuan, Li & Yu (2019)* proposed a label noise query and correction algorithm based on active learning to improve the quality of text annotation. *Wijeratne & Heravi (2014)* proposed a noise data filtering method based on keywords, which plays a good role in data preprocessing on Twitter text but has not been applied to other datasets. Due to the unique data characteristics of danmaku text, the text contains many popular words and emojis, making the mining and analysis of danmaku data more complex than traditional text content. Through the study of Danmaku data, key information that users pay attention to can be obtained, and related research results have been well applied in video recommendations. *Chen et al. (2020)* proposed the "TSCREC" model, which uses a bidirectional recursive unit to capture the semantics of existing comments. It provides some suggestions for users when they post danmaku comments. On the one hand, it enhances the real-time interaction between users in the video and encourages users to post more comments. The research on danmaku data has also been applied to video recommendation. By improving the traditional K-means clustering algorithm, *Hong et al. (2018)* studied the similarity between danmaku content and analyzed the association among users, which could understand the danmaku users and attempt the potential relationship among various videos but also could recommend more suitable video
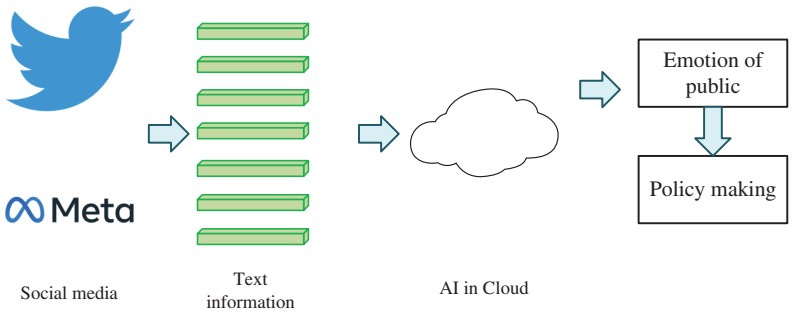

**Figure 1 Natural disaster public opinion analysis system.**

resources for viewers. *Trivedi & Patel (2022)* propose a hybrid work model to understand the public sentiments and achieve the best performance comparing with others. *Shi et al. (2022)* research the online opinion on the Weibo and analyze the change of opinion during the COVID-19. *Wu & Deng (2022)* propose a public opinion analysis framework, which can withdraw the information from the Internet.

## PUBLIC OPINION ANALYSIS SYSTEMS AND APPLICATIONS FOR NATURAL DISASTERS

In natural disaster public opinion analysis, the public sentiment feature extraction algorithm based on social media transmission can help the government and rescue agencies quickly understand the public sentiment and attitude to formulate response measures better. As shown in Fig. 1, this algorithm usually includes the following steps: 1. Data collection: Collect data related to natural disasters on social media platforms, including text, pictures, videos, *etc.*, which can be realized using technologies such as crawlers. 2. Text preprocessing: The collected text data is preprocessed, including word segmentation, stop words removal, part-of-speech tagging, named entity recognition and other processing, to facilitate subsequent sentiment analysis. 3. Sentiment analysis: Using sentiment analysis algorithm to analyze the sentiment of preprocessed text data, usually using sentiment dictionary or machine learning and other technologies to classify the sentiment of text data into positive, negative, neutral, *etc.* 4. Feature extraction: extract the key features in the sentiment analysis results, such as sentiment words, sentiment intensity, sentiment polarity, *etc.* 5. Visual analysis: Visualize the results of feature extraction, such as word cloud, heat map, timeline and other forms to show the changing trend of public sentiment and attitude.

The public emotion feature extraction algorithm based on social media communication can help the government and rescue agencies better understand the public emotion and attitudes, adjust the response strategy in time, and improve the response efficiency. For example, in a natural disaster event, the government and rescue agencies can use this algorithm to understand the public's attention to the disaster, satisfaction with the government and rescue agencies, and evaluation of disaster response measures, to better formulate response measures and improve public satisfaction.

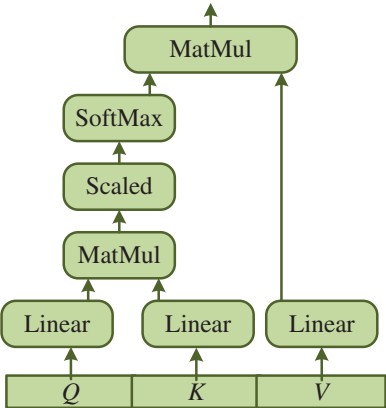

**Figure 2** **The structure of self-attention.** 

## Public emotion feature extraction algorithm based on social media communication

Following the above steps, we collect the public's social media information, such as blog text or audio and video, through sensing devices and Internet technology when natural disasters occur. The Transformer network (*Lian, Liu & Tao, 2021*) is employed to achieve features from the collected text and audio and video information as shown in Fig. 2.

First, we compute the self-attention mechanism. The public opinion information is quantized into feature F and the output and weight matrix $W^Q$, $W^K$, $W^V$ are calculated respectively to increase the association between features and enhance the universality of features, as shown below:

$$Q = FW^Q \tag{1}$$
$$K = FW^K \tag{2}$$
$$V = FW^V \tag{3}$$

Then, the output features Z of the self-attention layer are calculated, where $d_k$ is the dimension of the model.

$$Z = \text{Softmax}\left(\frac{QK^T}{\sqrt{d_k}}\right)V \tag{4}$$

Secondly, the output matrix $Z_{mul}$ of the multi-head self-attention layer is calculated, where H refers to the counting of attention heads, $Z_i$ denotes the $(i+1)$-th head. [...] means concatenating the H attention heads, $W^O$ refers to the extra matrix. Note that the dimension of $Z_{mul}$ and X are the same. The summation and layer-normalized are performed, LN represents layer normalization.

$$Z_{mul} = [Z_0, Z_1, \ldots, Z_i]W^O \tag{5}$$
$$Z_{mul} = LN(Z_{mul} + F) \tag{6}$$

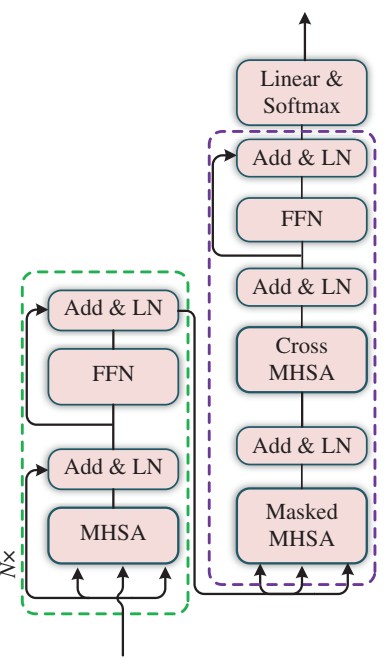

**Figure 3 The structure of Transformer.**     

Finally, $Z_{mul}$ is passed to the feedforward neural network, after which the summation and LN have performed again, as shown in Fig. 3.

A Transformer block contains a multi-head self-attention that includes the LN, self-attention, residual connection and multi-layer perceptron (MLP). The calculation process is as follows:

$$\hat{z}_i = MSA(LN(z_{i-1})) + z_{i+1} \tag{7}$$
$$z_i = MLP(LN(\hat{z}_i)) + \hat{z}_i \tag{8}$$
$$\hat{z}_{i+1} = MSA(LN(z_i)) + z_i \tag{9}$$
$$z_{i+1} = MLP(LN(\hat{z}_{i+1})) + \hat{z}_{i+1} \tag{10}$$

## Intelligent recognition of public emotional information

We regard short texts with different lengths as matrix input. Then, we employ multiple convolution kernels to extract the information on emotional features for the final classification. The model is composed of four parts: (1) Input layer module. It contains convolution, pooling and a fully connected layer. They represent all the emotional features as a vector. We can obtain an embedding matrix M, where each row of M represents a word vector. This M can be static, that is, fixed or non-static, that is constantly updated according to backpropagation. The length of the vector determines the width of the CNN kernel. The height of the CNN kernel generally uses multiple values to obtain richer feature expressions. In this article, the height of the CNN kernel uses 2, 3, 4, 5, 6, and 7 values, and the convoluting operation is conducted on every window of the emotion feature to achieve the features. Assuming that the width of the convolution kernel d is w and the height is h, a

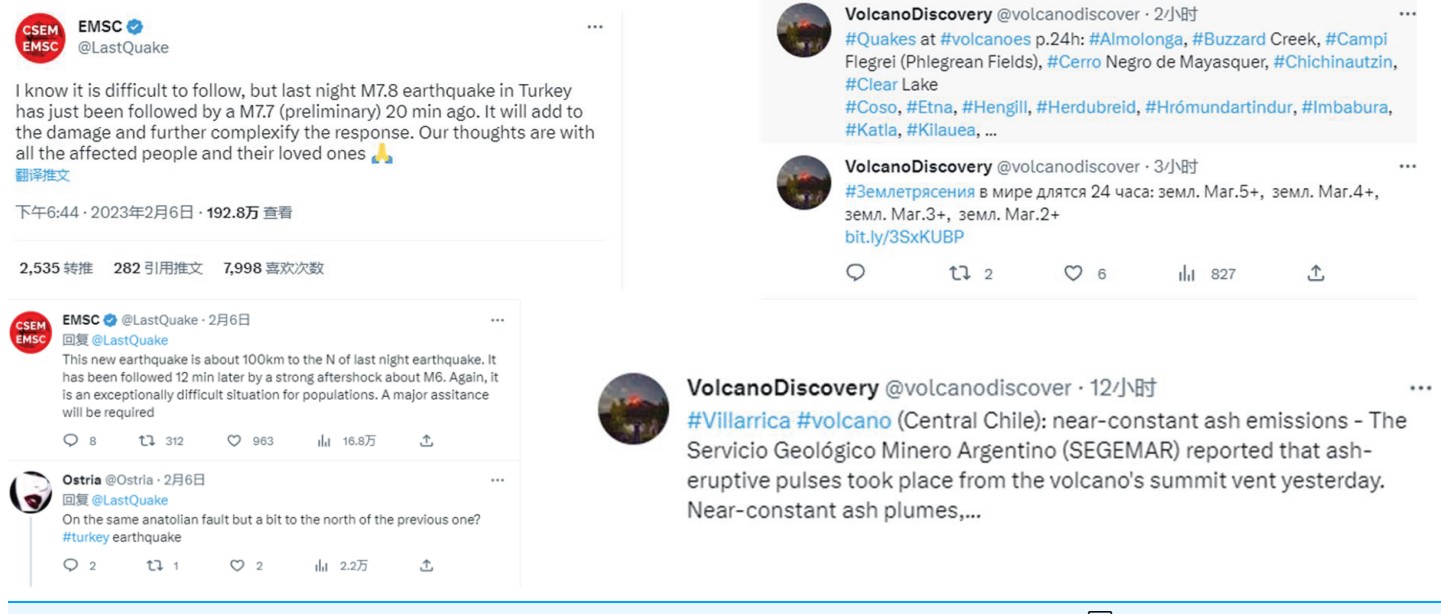

**Figure 4  The examples in the dataset.**

vector matrix can be obtained, which represents the row i to row j of the vector. Then the convolution operation can be expressed as follows. $A \in R_{s \times d}A[i:j]$

$$o_i = d \cdot A[i:i-h+1], \; i = 1,2,\ldots,s-h-1 \tag{11}$$

Add the upper bias b, after using the f activation, to get the desired feature. The equation is as follows:

$$c_i = f(o_i + b) \tag{12}$$

For a convolution kernel, s-h+1 features can be obtained, and a convolution operation can obtain the feature map

$$C = [c_1, c_2, \ldots, c_{s-h+1}] \tag{13}$$

The features obtained by CNN kernels with various sizes are not the same. The feature maps obtained after the convolution layer's convolution can be obtained using the downsampling function. Finally, the maximum downsampling is applied to get the final feature vector. The final features are input into the fully connected and softmax layers, and the dropout function is employed to prevent overfitting. The output is to calculate the probability values of the text in different categories to obtain the classification results.

## EXPERIMENT AND ANALYSIS

### Dataset and implement details

We use the Twitter dataset (https://figshare.com/articles/dataset/tweets_csv_gz/3465974/2) to evaluate the effectiveness of our method. We extracted news about natural disasters
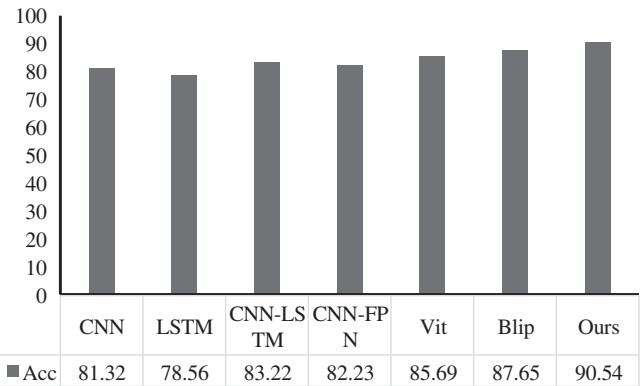

**Figure 5 Comparison with other methods.**

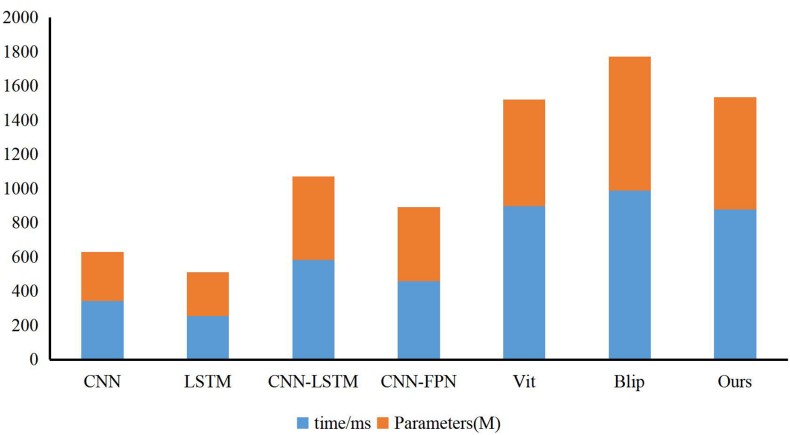

**Figure 6 The results of parameters, cost time and GFLOPs of the methods.**

and their corresponding public opinion comments from this dataset, and the sample of public opinion comments is shown in Fig. 4.

The experiments are carried out on a device with i7-13700 Cpu and Rtx 4090 Gpu, the operating system is Linux, and the network model is implemented under the TensorFlow framework. We set the total number of rounds of the experiment to 100. Besides, we regard the batch size as 32 and the initial learning rate as 0.01. Adam is used as the optimizer of the model, the momentum was 0.9, and the weight decay term was set to.$1 \times 10^{-4}$

To prove the performance of the method, we apply accuracy as the evaluation criterion, which is calculated as follows:

$$Accuracy = \frac{TP}{TP + FP} \tag{14}$$

where TP represents the number of samples predicted correctly, FP is the number of incorrectly predicted samples.

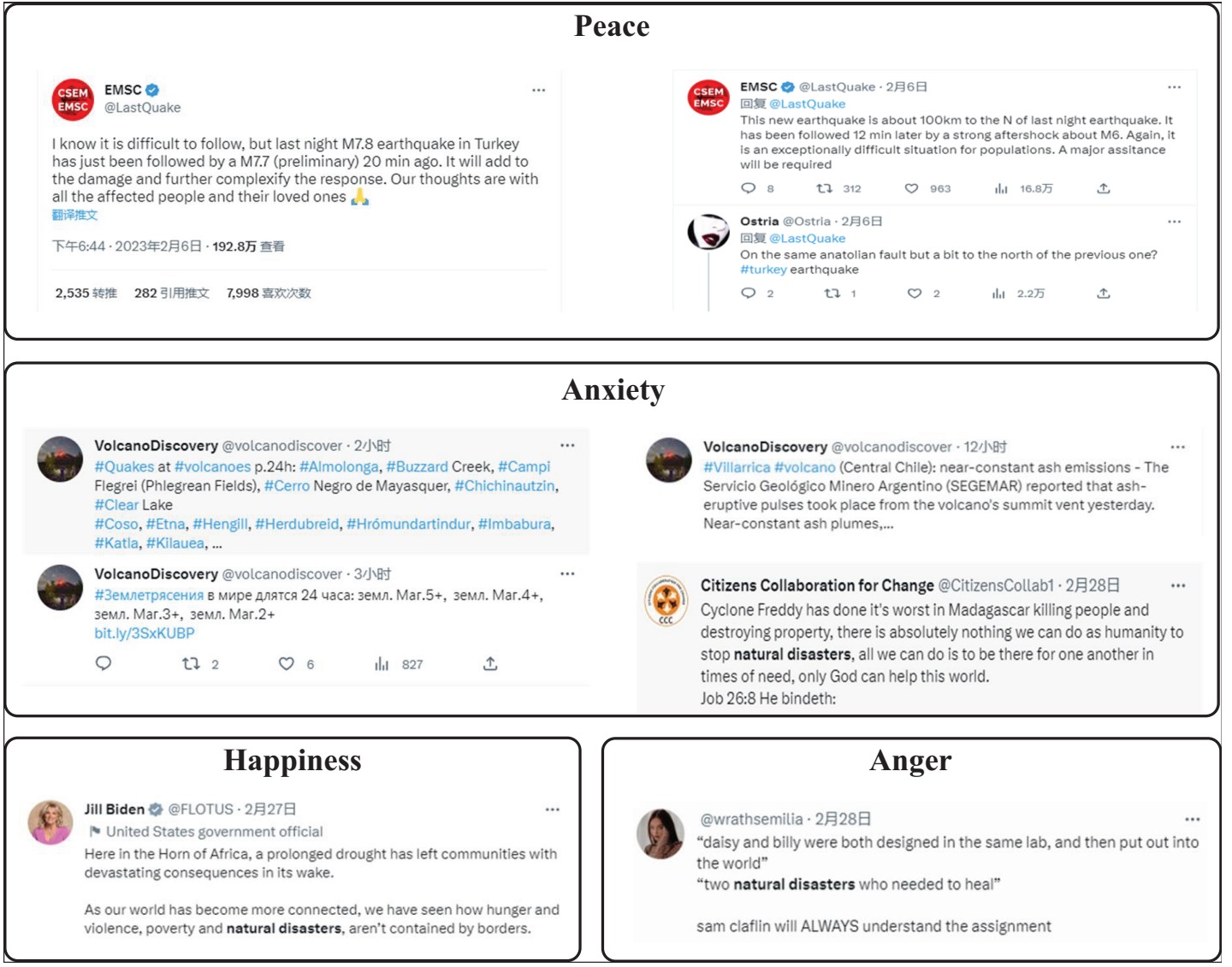

**Figure 7** The results of our methods about pre-lessons.

## Results and discussion

We conducted experiments on the Twitter dataset to extract public emotion features based on social media communication. At the same time, we set four emotion levels: anxiety, anger, happiness, and calm. Compared with five advanced technologies: CNN, LSTM, CNN-LSTM, CNN-FPN, Vit, and Blip, we present the results in Fig. 5. Our method achieves excellent teaching plan content score classification performance, with an accuracy of 90.54%. Compared with the two convolutional models of CNN and CNN-FPN, the accuracy evaluation index is increased by 9.22% and 8.31%. While comparing with the LSTM-based model, our method increases the accuracy by 11.98% and 7.32%, respectively. In addition, compared with the two transformer-based methods, Vit and Blip, the accuracy

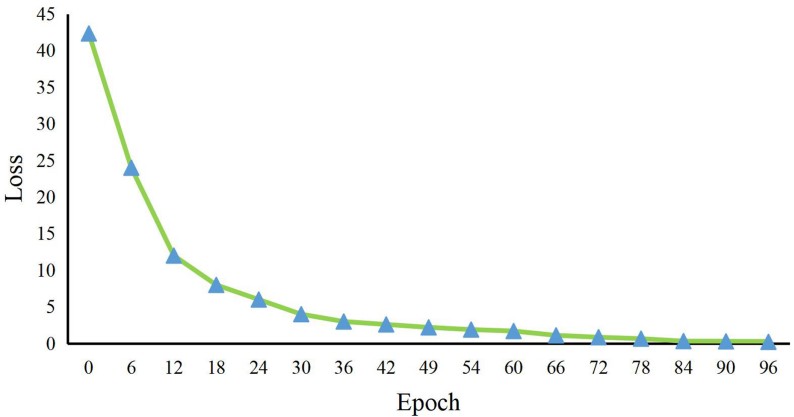

**Figure 8** The training of our methods.

**Table 1** Test of our application.

| Volunteer | 1 | 2 | 3 | 4 | 5 | 6 | 7 | 8 | 9 | 10 | System |
|---|---|---|---|---|---|---|---|---|---|---|---|
| Earthquake | 1 | 0 | 3 | 2 | 1 | 0 | 1 | 1 | 1 | 1 | 1 |
| Flood | 2 | 3 | 2 | 0 | 2 | 2 | 2 | 2 | 2 | 3 | 2 |
| Drought | 0 | 0 | 0 | 2 | 1 | 0 | 0 | 3 | 1 | 0 | 0 |
| Volcanic eruption | 1 | 2 | 3 | 3 | 3 | 3 | 1 | 0 | 3 | 3 | 3 |
| Forest fire | 1 | 1 | 1 | 1 | 0 | 1 | 2 | 0 | 1 | 1 | 1 |

is improved by 4.85% and 2.89%, respectively. Therefore, from the quantitative analysis, the performance of our method can surpass other methods, reaching the level of Sota.

As shown in Fig. 6, our model has a larger number of parameters and a longer inference time compared with CNN, LSTM, CNN-LSTM, CNN-FPN, Vit, and Blip methods. However, considering that the CNN-based model quickly loses the details of emotional features, the LSTM-based model fails some global features and has the problem of inaccurate classification. Based on ensuring accuracy first, our model appropriately cuts the model's structure, providing sensitivity and preventing the loss of features. Finally, we show some public opinion sentiment classification made by our method, as shown in Fig. 7. Our proposed public sentiment feature extraction algorithm based on social media transmission can accurately identify the public opinion status in natural disasters. In addition, we also show the training duration in Fig. 8.

## Application test

To verify the application of this method in real scenarios, we recruit 100 universities to comment on five recent natural disasters and simulate the generation of public opinion. Then, we use our system to evaluate the generated simulated public opinion and obtain the public attitude and public opinion recognition results for a series of events after natural disasters. The results are shown in Table 1. We randomly selected a part of them for display, where 0, 1, 2, and 3 represent the four emotional levels of anxiety, anger, happiness, and calm, respectively.

We can find that for each kind of natural disaster, our system can accurately identify the main opinion in the public opinion to realize the analysis and understanding of the public opinion, which provides good information for making policy plans.

## CONCLUSION

In order to analyze the content and development trend of public opinion in natural disasters, this article discusses integrating social media platforms and public opinion of natural disasters and proposes a public sentiment feature extraction algorithm based on social media transmission. The purpose of collecting public emotional expression information in natural disasters is achieved by proposing a feature extraction algorithm based on Transformer technology to sense the emotion of public opinion information in social media. Then, the public's overall emotional state was evaluated by recognizing the emotional expression information. Finally, this evaluation information summarizes the public's demands and needs in natural disasters. A public opinion analysis system based on social media communication is constructed. Experiments show that the algorithm proposed in this article can accurately identify the public emotional state in social media communication. The system built in this article can realize the end-to-end deconstruction of the status of natural disasters and their impact on the public.

### Funding
This work was supported by the Fundamental Research Funds for the Central Universities (No. ZY20215146). The funders had no role in study design, data collection and analysis, decision to publish, or preparation of the manuscript.

### Grant Disclosures
The following grant information was disclosed by the authors:
Fundamental Research Funds for the Central Universities: ZY20215146.

### Competing Interests
The authors declare that they have no competing interests.

### Author Contributions
- Shanshan Li conceived and designed the experiments, performed the experiments, analyzed the data, performed the computation work, prepared figures and/or tables, authored or reviewed drafts of the article, and approved the final draft.
- Xiaoling Sun conceived and designed the experiments, performed the experiments, analyzed the data, performed the computation work, prepared figures and/or tables, authored or reviewed drafts of the article, and approved the final draft.

### Data Availability
The code are available in the Supplemental File.

The dataset is available at Figshare: Quezada, Mauricio; jkalyana@ucsd.edu; bpoblete@dcc.uchile.cl; gert@ece.ucsd.edu (2016): Twitter News Dataset. figshare. Dataset. https://doi.org/10.6084/m9.figshare.3465974.v2.

## Supplemental Information

Supplemental information for this article can be found online at http://dx.doi.org/10.7717/peerj-cs.1417#supplemental-information.

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
