# Peer review of "Application of public emotion feature extraction algorithm based on social media communication in public opinion analysis of natural disasters"

_PeerJ Computer Science, doi:10.7717/peerj-cs.1417_

## Round 0.1 · original submission · Major Revisions

Dear authors

Your paper has been reviewed by me and the experts, and although it has merits on technical grounds, the experts have listed some improvements before we further consider it further. Therefore please revise the paper carefully in light of these suggestions to improve the quality of your manuscript.
Thanks.

Reviewer 1 ·

Basic reporting

This paper proposes a public opinion analysis method based on emotional characteristics, which uses feature extraction and Transformer technology to perceive emotions in public opinion samples. Then, through the extracted features, the public emotions are intelligently identified to realize the collection of public emotions in natural disasters. Finally, the public's demands and needs in natural disasters can be obtained through the collected emotional information, and a natural disaster public opinion analysis system based on social media communication can be realized. It can deconstruct the status quo of natural disasters point by point and grasp the situation of disasters in real time.

1) The author's language expression needs to be improved, and there are some incomplete contents in the paragraphs of the article that need to be improved.
(2) The content of the research keyword part needs to be modified. There are too few quoted words in this part, which can be supplemented appropriately.
(3) In terms of the impact of social media on the analysis of natural disaster events, more references need to be added, and Reference[4] is not enough;
(4) The last part of the introduction lacks an introduction of the main technical means;
(5) Similarly, in this part, I failed to understand the methodological innovation and contribution of this research.
(6) The literature review needs to be adjusted, supplemented or replaced with some studies in the recent 3 years to make them more representative.

Experimental design

(1) In Section 3.1, more references need to be added, especially for formulas (7) ~(10);
(2) In Section 3.2 Intelligent recognition of public emotional information, I would like to see more specific introduction;

Validity of the findings

(1) Why is Accuracy only used as an evaluation index?

Additional comments

Please have someone competent in the English language and the subject matter of your paper go over the paper and correct it.

Reviewer 2 ·

Basic reporting

In order to analyze the content and development trend of public opinion in natural disasters. This paper discusses how to integrate social media platforms and public opinion about natural disasters, and proposes a public emotion feature extraction algorithm based on social media communication. By proposing a feature extraction algorithm based on Transformer technology for emotional perception of public opinion information in social media, the purpose of collecting public emotional expression information in natural disasters can be achieved. After careful readin,g the authors need a revision of the paper. The presentation of the paper needs a professional improvement.

1. Please sort out the article again and summarize 4-5 representative keywords. All keywords should be written in alphabetical order.
2. There are few explanations of the rationale for the study design.
3. Try to set the problem discussed in this paper in more clear, write one section to define the focused problem.
4. There are just descriptions of other papers but not a real analysis applied to this proposed method.
5. What is the originality of this research? Motivation should also be clearly highlighted.
6. Improve this paragraph, the paper research gap and originality should be better presented at the end of the introduction section.
7. Findings should be contextualized in the literature and should be explicit about the added value of the study towards the literature.
8. Check target journal/ institution for prescribed format. Moreover, double-check spelling, punctuation marks, and titles for accuracy in copying the texts verbatim.
9. All references should be written according to the journal referencing style.

Experimental design

1. The investigation has been conducted rigorously.
2. Methods are described with sufficient information to be reproducible.

Validity of the findings

1. There are just descriptions of other papers but not a real analysis applied to this proposed method.
2. Originality and novelty of this research is not clear. Motivation should also be clearly highlighted.
3. Improve this paragraph, the paper research gap and originality should be better presented at the end of the introduction section.
4. Findings should be contextualized in the literature and should be explicit about the added value of the study towards the literature.

Additional comments

1. Please sort out the article again and summarize 4-5 representative keywords. All keywords should be written in alphabetical order.
2. There are few explanations of the rationale for the study design.
3. Try to set the problem discussed in this paper in more clear, write one section to define the focused problem.
4. There are just descriptions of other papers but not a real analysis applied to this proposed method.
5. What is the originality of this research? Motivation should also be clearly highlighted.
6. Improve this paragraph, the paper research gap and originality should be better presented at the end of the introduction section.
7. Findings should be contextualized in the literature and should be explicit about the added value of the study towards the literature.
8. Check target journal/ institution for prescribed format. Moreover, double-check spelling, punctuation marks, and titles for accuracy in copying the texts verbatim.
9. All references should be written according to the journal referencing style.

---

## Round 0.2 · accepted · Accept

Thank you for your fine contribution to our esteemed journal. Good luck

Reviewer 1 ·

Basic reporting

All suggested changes have been incorporated properly

Experimental design

All suggested changes have been incorporated properly

Validity of the findings

All suggested changes have been incorporated properly

Additional comments

All suggested changes have been incorporated properly

Reviewer 2 ·

Basic reporting

The current version of the paper presents an expressive improvement as compared to the previous one.
The authors provided acceptable answers to all my questions and no more issues were detected in the presented manuscript. Therefore, I recommend the acceptance of the paper in its current form.

Experimental design

The current version of the paper presents an expressive improvement as compared to the previous one.
The authors provided acceptable answers to all my questions and no more issues were detected in the presented manuscript. Therefore, I recommend the acceptance of the paper in its current form.

Validity of the findings

The current version of the paper presents an expressive improvement as compared to the previous one.
The authors provided acceptable answers to all my questions and no more issues were detected in the presented manuscript. Therefore, I recommend the acceptance of the paper in its current form.

Additional comments

The current version of the paper presents an expressive improvement as compared to the previous one.
The authors provided acceptable answers to all my questions and no more issues were detected in the presented manuscript. Therefore, I recommend the acceptance of the paper in its current form.